# Guidelines for Fabricating Highly Efficient Perovskite Solar Cells with Cu_2_O as the Hole Transport Material

**DOI:** 10.3390/nano12193315

**Published:** 2022-09-23

**Authors:** Sajid Sajid, Salem Alzahmi, Imen Ben Salem, Ihab M. Obaidat

**Affiliations:** 1Department of Chemical & Petroleum Engineering, United Arab Emirates University, Al Ain P.O. Box 15551, United Arab Emirates; 2National Water and Energy Center, United Arab Emirates University, Al Ain P.O. Box 15551, United Arab Emirates; 3College of Natural and Health Sciences, Zayed University, Abu Dhabi P.O. Box 144534, United Arab Emirates; 4Department of Physics, United Arab Emirates University, Al Ain P.O. Box 15551, United Arab Emirates

**Keywords:** numerical simulation, inorganic charge-transporting layer, performance optimization, perovskite solar cell

## Abstract

Organic hole transport materials (HTMs) have been frequently used to achieve high power conversion efficiencies (PCEs) in regular perovskite solar cells (PSCs). However, organic HTMs or their ingredients are costly and time-consuming to manufacture. Therefore, one of the hottest research topics in this area has been the quest for an efficient and economical inorganic HTM in PSCs. To promote efficient charge extraction and, hence, improve overall efficiency, it is crucial to look into the desirable properties of inorganic HTMs. In this context, a simulation investigation using a solar cell capacitance simulator (SCAPS) was carried out on the performance of regular PSCs using inorganic HTMs. Several inorganic HTMs, such as nickel oxide (NiO), cuprous oxide (Cu_2_O), copper iodide (CuI), and cuprous thiocyanate (CuSCN), were incorporated in PSCs to explore matching HTMs that could add to the improvement in PCE. The simulation results revealed that Cu_2_O stood out as the best alternative, with electron affinity, hole mobility, and acceptor density around 3.2 eV, 60 cm^2^V^−1^s^−1^, and 10^18^ cm^−3^, respectively. Additionally, the results showed that a back electrode with high work-function was required to establish a reduced barrier Ohmic and Schottky contact, which resulted in efficient charge collection. In the simulation findings, Cu_2_O-based PSCs with an efficiency of more than 25% under optimal conditions were identified as the best alternative for other counterparts. This research offers guidelines for constructing highly efficient PSCs with inorganic HTMs.

## 1. Introduction

Perovskite solar cells (PSCs) utilizing organic–inorganic halide perovskites have been realized as emerging solar cells, with rapidly progressing power conversion efficiencies (PCEs), ease of processing, and relatively low-cost production, among other photovoltaics [1,2]. The first use of organic–inorganic halide perovskites was reported in dye-sensitizer solar cells with a PCE of 3.9% in 2009, which was a huge achievement in the field of excitonic solar cells [3]. Following that, perovskite quantum dots were utilized in PSCs, which had a 6.5% PCE with low stability due to liquid electrolytes [4]. Then, 2,2′,7,7′-Tetrakis[N,N-di(4-methoxyphenyl)amino]-9,9′-spirobifluorene (spiro-OMeTAD) was used as a hole transport material (HTM), which increased the stability of PSCs in 2012, achieving a PCE of 9.1% [5]. The PCE of PSCs was improved further by modifying the halide mixture, deposition techniques, and HTMs in organic–inorganic halide perovskites, resulting in a PCE of 22.1% in 2016. Subsequently, most of the PSCs with significant increases in PCE over 25% have been reported with organic HTMs [6]. High-performance organic HTMs, however, are frequently made of costly organic conjugated molecules, such as spiro-OMeTAD or semiconductor polymers [7,8], which restricts their potential to be manufactured at a large scale and at a reasonable cost [9,10,11,12]. In addition, hygroscopic additives, such as lithium or cobalt salts, are frequently needed to improve their hole mobilities and appropriate Fermi level alignment, resulting in device instability through water-assisted photoactive material degradation [9,13].

In this context, alternatives such as NiO, Cu_2_O, CuI, CuSCN, molybdenum trioxide (MoO_3_), and copper zinc tin sulfide (CuZnSnS_2_) have emerged as low-cost, highly conductive, stable, and dopant-free HTMs [14,15,16,17,18,19,20]. Furthermore, the energy levels of inorganic HTMs are remarkably similar to perovskite’s frontier energy levels. MoO_3_ HTMs employed in PSCs displayed a high open-circuit voltage (V_oc_) of 1.15 V [21]. Then, the use of NiO HTM attained a PCE of 7.6% [22]. Because of the lower fill factor (FF), the performance of NiO-based PSCs was still inferior to that of organic HTMs due to the low conductivity feature of NiO. Inorganic CuO HTMs were further investigated as an alternative to organic HTMs and attained a PCE of 19.0% in PSCs with good stability [23]. The PCEs of inverted PSCs using inorganic HTMs have been reported to be around 10%; however, this figure has lately risen to more than 20% [24]. There is still a significant difference between the PCEs of PSCs in particular regular design-employing inorganic HTMs and their organic counterparts. Hence, it is critical to examine the factors that influence PSC performance.

Nevertheless, due to perovskite’s incompatibility with a variety of chemical precursors and deposition techniques, research has primarily focused on inverted structures of PSCs using inorganic HTMs. Thus far, very few investigations have looked into inorganic HTMs that can be used for cost-effective regular PSC architectures [25,26]. The use of copper derivatives (copper sulfide (CuS), copper chromium oxide (CuCrO_2_), and Cu_2_O) in regular PSCs has showcased potential deposition techniques of inorganic HTMs [27,28,29]. Cu_2_O has recently been used as an HTM in the regular design of PSCs, which has resulted in PCEs of 17% to 18%, demonstrating the potential of this HTM for further PCE enhancement [29,30]. In this context, theoretical studies are just as vital as experimental ones [31,32]. Therefore, a simulation study is conducted to provide guidelines for taking into account crucial requirements for producing highly efficient regular PSCs based on Cu_2_O as HTMs. With the help of simulations and optimization, a 25.2% PCE of Cu_2_O-based PSCs is produced under optimal conditions. When fabricating PSCs using inorganic HTMs, the simulation findings imply that electron affinity, hole mobility, acceptor density, and metal work function should all be considered.

## 2. Simulation Parameters and Computational Details

Simulation methods can help explain the phenomena existing in solar cells [33,34,35,36,37]. In order to gain a better understanding, a theoretical study using SCAPS software was conducted. A planar PSC configuration (FTO/TiO_2_/MAPbI_2_Br/HTM/Au) was simulated as a baseline to evaluate the impact of our method, as depicted in Figure 1a,b. The basic physical parameters for the simulation are listed in the Table 1. Most of them were chosen from recent publications [38,39,40]. The solar radiation spectrum of AM1.5G was used in this simulation as a light source. The light reflection was set to 0 and 1 for the top and bottom contacts, respectively. With a Gaussian energy distribution, defects in the energy levels were considered in the middle of their characteristic energy bandgap of 0.1 eV in the simulated thin films.

Modeling techniques define fundamental phenomena in photovoltaic devices, which allows for the instinctive classification of optimized operating conditions for each parameter [41]. For further practical applications of heterojunction solar cells, SCAPS is ideal for intraband tunneling and trap-assisted tunneling [42]. The strength of this modeling technique was initially simulated using various inorganic HTMs in the PSCs, and later, the results were compared with experimental results [30,39]. For holes and electrons, capture cross-sections were set to 10–16 cm^2^, which resulted in perovskites having carrier diffusion lengths of approximately 1 µm [43]. Active layer thickness was set to 400 nm in order to provide the efficient collection of charges. The recombination rate of the bimolecular electron-hole has a tremendous effect on the solar cell efficiency, so the total recombination in the as-simulated PSCs was taken into account by SCAPS. SCAPS software does not take into account the recombination of charges at the corresponding interfaces. Therefore, two thin defect layers were stacked at the HTM–perovskite and electron transport material (ETM)–perovskite interfaces to overcome the interfacial defect density, as presented in Table 2. This structure allowed for the evaluation of the role of interface adjustment in the simulated and experimental effects in determining the device efficiency [44,45].

**Table 1 nanomaterials-12-03315-t001:** Basic parameters of the materials used for as-simulated PSCs.

Parameters/Units	TiO_2_ [46]	MAPbI_2_Br [47,48,49]	CuSCN [50]	CuI [51,52]	Cu_2_O [53]	NiO [11]
Thickness/nm	30	400	80	80	80	80
Defects/cm^3^	10^17^	10^17^	10^17^	10^17^	10^17^	10^17^
Band gap/eV	3.2	1.8	3.4	3.1	2.17	3.6
Electron affinity/eV	4.0	3.93	1.7	2.1	3.2	1.46
Dielectric constant	100	25	10	6.5	6.6	11.2
Effective valence band density/cm^−3^	2 × 10^20^	3 × 10^18^	2.5 × 10^19^	2.5 × 10^19^	2.5 × 10^19^	2.5 × 10^19^
Effective conduction band density/cm^−3^	10^21^	4 × 10^18^	1.5 × 10^18^	1.5 × 10^18^	1.5 × 10^18^	1.5 × 10^18^
Electron mobility/cm^2^·V^−1^·s^−1^	0.006	15	0.0002 [54]	1.5 [12]	0.02 [55]	1.4
Holemobility/cm^2^·V^−1^·s^−1^	0.003	15	0.2 [54]	4.8 [12]	90 [52,55]	4.9
Donor concentration/cm^−3^	5 × 10^19^	1 × 10^18^	0	0	0	0
Acceptor concentration/cm^−3^	0	1 × 10^18^	3 × 10^18^	3 × 10^18^	3 × 10^18^	3 × 10^18^

**Table 2 nanomaterials-12-03315-t002:** Basic parameters for defect layers at the interfaces of ETM–perovskite and HTM–perovskite [11,12,56].

Parameters and Units	ETM–Perovskite	HTM–Perovskite
Dielectric constant	30	6.6
Band gap/eV	1.8	2.17
Electron affinity/eV	3.93	3.2
Thickness/μm	0.002	0.002
Electron and hole mobility/cm^2^·V^−1^·s^−1^	50, 50	0.5, 0.5
Acceptor concentration/cm^−3^	0	2 × 10^17^
Donor concentration/cm^−3^	2.14 × 10^17^	0
Effective conduction band density/cm^−3^	2.5 × 10^20^	2 × 10^17^
Effective valence band density/cm^−3^	2.5 × 10^20^	1.1 × 10^19^
Characteristic energy for donor- and acceptor-like tails/eV	0.015, 0.015	0.01, 0.01
Band tail density of states/cm^−3^eV^−1^	1 × 10^14^	1 × 10^14^
Capture cross-section for electrons and holes in donor tail states/cm^2^	1 × 10^−15^, 1 × 10^−17^	1 × 10^−15^, 1 × 10^−17^
Capture cross-section for electrons and holes in acceptor tail states/cm^2^	1 × 10^−17^, 1 × 10^−15^	1 × 10^−17^, 1 × 10^−15^
Switch-over energy/eV	0.7	0.8
Density of mid-gap acceptor- and donor-like state/cm^−3^eV^−1^	1 × 10^16^ to 1 × 10^19^	1 × 10^17^ to 1 × 10^19^
Capture cross-section of electrons and holes in donor mid-gap states/cm^2^	1 × 10^−17^, 1 × 10^−18^	1 × 10^−16^, 1 × 10^−17^
Capture cross-section of electrons and holes in acceptor mid-gap states/cm^2^	1 × 10^−18^, 1 × 10^−17^	1 × 10^−17^, 1 × 10^−16^

## 3. Results and Discussion

### 3.1. Identifying the Best HTM

In order to identify the best HTM for efficient PSCs, the photovoltaic parameters with different HTMs, such as NiO, Cu_2_O, CuI, and CuSCN, were simulated. The schematic diagram of the simulated devices and the energy band diagrams are given in Figure 1a,b. The basic parameters of the used materials are given in Table 1. While modeling for various HTMs, the parameters of the ETM and perovskite layers were fixed. Additionally, the same values for both bulk and interface defect densities were used to maintain a consistent analysis.

The current–voltage (J-V) characteristic curves and external quantum efficiency (EQE) of the as-simulated PSCs are depicted in Figure 2a,b, respectively. The photovoltaic metrics of the as-simulated devices are presented in Table 3. The simulation results showed that, when the HTM parameters (mentioned in Table 1) were altered, the photovoltaic metrics varied, particularly the Voc and fill factor (FF), as shown in Figure 2a and Table 2. Furthermore, it can be seen from Figure 2b that there was a little variance in the EQEs of the as-simulated PSCs. From the simulation results, it was noticed that the PSC with the Cu_2_O HTM delivered a higher performance compared to the other HTMs. The superior performance of the Cu_2_O-based PSC was attributed to the alignment of frontier energy levels, suitable electron affinity, and hole mobility of the as-simulated materials, as discussed below. Additionally, the simulation results closely match the experimental findings of previous published research [29,30].

For the efficient flow of electrons from perovskite to the ETM, the conduction band of the ETM should align with the perovskite’s conduction band [57,58]. It is also essential that the valance band of the HTM match with the absorber layer in order to facilitate the transport of holes between them [59,60]. In other words, this can increase the built-in potential, which is critical for improving the V_oc_ values of the PSCs. To observe the energy band matching between HTMs and the perovskite layer, the energy band levels of the as-simulated PSCs were analyzed, as shown in Figure 3. It can be seen from the energy levels of the as-simulated PSCs that there was a formation of an energy barrier in the cases of the NiO, CuI, and CuSCN HTMs. This energy barrier hindered the transportation of holes from the perovskite layer to the HTM, and as a result, the performances of the PSCs dropped. Energy barriers with values of −0.48 eV, −0.40 eV, −0.21 eV, and −0.11 eV were calculated for NiO, CuSCN, CuI, and Cu_2_O, respectively. Since NiO had the highest energy barrier, therefore the lowest V_oc_ of 0.72 V and lowest PCE of 8.5% were obtained. On the other hand, the Cu_2_O HTM had the lowest energy barrier of −0.11 eV to the flow of holes, and thus, the PSC delivered the highest PCE of 25.2%. With regard to the low energy barrier resulting from frontier energy level alignment with the perovskite layer, Cu_2_O had a superior performance. Cu_2_O was, therefore, regarded as the best HTM for usage in the PSCs. In the following sections, we examine and analyze various factors of Cu_2_O that affect the performance of PSCs.

### 3.2. Electron Affinity of Cu_2_O and PSC Performance

The PCE of a PSC can be improved by ensuring that the HTM and perovskite layer are aligned in terms of frontier energy levels. In this context, the valence energy band (E_v_) must be taken into account in order to understand this difference between the HTM and the perovskite layer. The mismatching between the E_v_ of perovskite and an HTM is generally denoted by valance band offset (VBO), which can be determined by the electron affinities of the corresponding materials. In addition, if the VBO between the HTM and perovskite layer is small, the performance of the PSC is superior compared to higher values of VBO. In the simulation, it was observed that the electron affinity of Cu_2_O had a significant impact on the performance of PSCs. There were essentially no differences in the patterns of photovoltaic parameters with varied electron affinities of HTM from those reported in a previous work [61]. In order to obtain the optimal value for the electron affinity of Cu_2_O, the affinity was varied from 1.5 eV to 3.7 eV, as depicted in Figure 4a,b. The major impacts obtained were on the V_oc_ and FF values of the PSCs. The V_oc_ and FF values were improved substantially when the affinity was changed from 2.0 eV to 3.2 eV. Correspondingly, the PCE was boosted from 19.6% to 25.2%.

The simulation results revealed that high or low values of electron affinity resulted in inferior device performance, which was due to mismatch the in E_v_ levels of HTM and perovskite. In order to achieve the highest PCE, we assumed that the optimal electron affinity for Cu_2_O was 3.2 eV, and that for perovskite was 3.93 eV. Furthermore, aligning the frontier energy levels was desirable to reduce the recombination rate at the interface between Cu_2_O and perovskite, as shown in Figure 4c. Consequently, the PCEs of the PSCs could be improved by manipulating the E_v_ of the HTM relative to the perovskite layer.

### 3.3. Hole Mobility of Cu_2_O and PSC Performance

Since the HTM is responsible for collecting and transporting holes from the perovskite layer to the back electrode, the hole mobility (μ_h_) of the used HTM should be high enough to transport holes before they recombine. An optimum value of hole mobility up to 90 cm^2^V^−1^s^−1^ for Cu_2_O was considered from the previous literature [61]. However, the μ_h_ of Cu_2_O was adjusted from 20 to 100 cm^2^V^−1^s^−1^ to examine the effect on the performance of the PSCs. The J-V curves and PCEs of the as-simulated PSCs as a function of the hole mobility of Cu_2_O are depicted in Figure 5. It can be seen that the short-current density (J_sc_) remained nearly constant, whereas the FF and V_oc_ improved as the μ_h_ of Cu_2_O was increased. Because of the improvement in hole conduction across the Cu_2_O HTM, the PCE increased from 15.4% to 25.2% when the μ_h_ was increased from 20 to 100 cm^2^V^−1^s^−1^. The improvement in PCE could be attributed to the high μ_h_ that reduced the series resistance within the PSCs [62]. It should be noted that the simulation results showed almost no changes in the PCEs for μ_h_ from 60 to 100 cm^2^V^−1^s^−1^, which is consistent with previous results [63,64]. The simulation results suggested that the optimal value of μ_h_ that corresponded to the highest PCEs was in the range from 60 to 100 cm^2^V^−1^s^−1^. These results provide guidelines for the optimization of PSC performance by adjusting the hole mobilities of HTMs.

### 3.4. Acceptor Density of Cu_2_O and PSC Performance

It was reported that p-type doping of an HTM produced more positive charges (majority charge carriers) and, thus, improved the bulk conductivity of the HTM and the performance of the PSCs [35]. Majority charge carriers that can be created at suitable acceptor densities (N_a_) can greatly boost the photovoltaic parameters of PSCs. When using PSCs, an iterative approach of doping concentration aids in the improvement of their overall performance [65]. In order to understand the influence of N_a_ on the performance of PSCs, the N_a_ values were changed from 10^7^ to 10^18^ cm^−3^ in the Cu_2_O. As a function of the N_a_ values of Cu_2_O, the J-V curves, V_oc_, FF, and PCE of the as-simulated devices are presented in Figure 6a–c, respectively. As the N_a_ of Cu_2_O increased, the V_oc_ increased from 1.13 to 1.29 V. It was the rise in the built-in electric potential at the Cu_2_O–perovskite interface that was responsible for the greater values of V_oc_ observed at higher N_a_ values. The FF and PCE increased from 76.6 to 83.6% and 20.2 to 25.2%, respectively, with the increase in N_a_ of Cu_2_O. The larger N_a_ value raised the electric potential, and therefore, the built-in electric field at the interface of the perovskite and HTM increased. The built-in electric field promoted the oriented charge carrier transportation and, thus, minimized the recombination losses [66]. In addition, the reduced recombination rate of charge carriers boosted the PCE of the PSCs by reinforcing the separation of charge carriers. From the simulations, the optimal value for N_a_, which resulted in the highest PCE, was 10^18^ cm^−3^. This optimal N_a_ value of Cu_2_O is consistent with earlier reports [47]. This was attributed to shallow coulomb traps, which increased the hole mobility of Cu_2_O.

### 3.5. Contact of Back Electrode with Cu_2_O and PSC Performance

Because the Cu_2_O layer was placed on top of the perovskite and the back electrode made contact with this HTM, it was critical to explore charge carrier transport when Ohmic or Schottky formations occurred, as these could alter charge collection [67]. The proper collection of holes through the back electrode requires the establishment of an Ohmic or Schottky contact with a small barrier. In this context, various metal electrodes, such as Ag, Cu, and Au with work functions of 4.74 eV, 4.90 eV, and 5.10 eV, respectively, were integrated in the as-simulated PSCs. It was revealed that the performance of the PSCs was affected by varying the work function value, as can be seen in Figure 7. In the presence of variable work function, the obtained behaviors of V_oc_, FF, and PCE matched those reported in previously published research [67]. It was revealed that a metal electrode with high work function resulted in an improved V_oc_ of 1.29 V. Furthermore, the FF climbed to a maximum value of 83.65% with rising work function. Due to the improved built-in voltage, the V_oc_ increased as the work function of the metal electrode increased. For the work function value of 5.1 eV, the PCE improved, reaching a maximum value of 25.21%. The improvement in the performance of PSCs could be attributed to the small series resistance resulting from a decrease in the Schottky barrier at the interfaces of Cu_2_O–HTM and Au–electrode. In Figure 7b,c, it is shown that Schottky barriers were formed in the energy band diagram for the work functions of electrodes of 4.74 eV and 4.90 eV, respectively. In Figure 7d, a Schottky barrier for holes was significantly reduced when the work function was equal to or smaller than the valence energy band of the HTM [68], resulting in a significant enhancement in the efficiency of PSCs. The findings of the simulation showed that Cu_2_O could be utilized in the PSCs, but the deposition of inorganic HTMs in the regular designs of PSCs should also be taken into consideration. For instance, inorganic HTMs are challenging to dissolve in the non-polar solvents needed to preserve the perovskite layer. In this regard, surfactant modifications [29] or chemical solvents, such as dipropyl sulfide [30] and isopropanol suspension [69], that are less aggressive to the degradation of the perovskite layer might be used to deposit solution-processed inorganic HTMs in regular PSCs.

## 4. Conclusions

A numerical analysis was performed to find the optimum conditions for PSCs with inorganic HTMs. Several factors that could affect the performance of PSCs were thoroughly investigated. According to the simulation results, the optimal electron affinity, hole mobility, and acceptor density of Cu_2_O were found to be 3.2 eV, 60 to 100 cm^2^V^−1^s^−1^, and 10^18^ cm^−3^, respectively. The simulation findings showed that a matched valance energy band of Cu_2_O resulted in improvements in the performance of the PSCs, whereas an unmatched valance energy band of Cu_2_O led to a high charge recombination rate and poor device performance. Low work function electrodes impeded charge transport by forming large Schottky barriers; hence, high work function of a metal electrode is needed for a low charge transport barrier. A PCE of 25.2% was attained under optimal conditions, demonstrating that Cu_2_O-based PSCs are promising for future applications.

## Figures and Tables

**Figure 1 nanomaterials-12-03315-f001:**
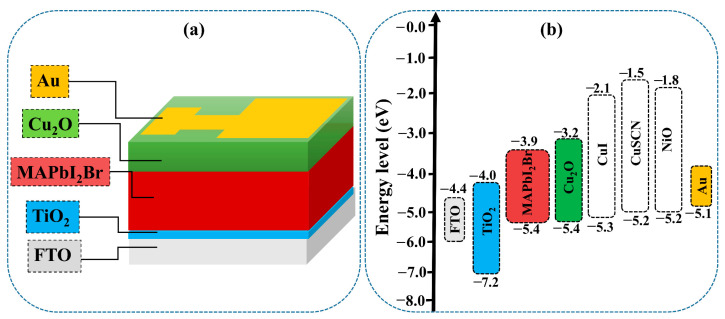
Schematic illustration of the as-simulated devices (**a**) and energy band diagram of the used materials (**b**).

**Figure 2 nanomaterials-12-03315-f002:**
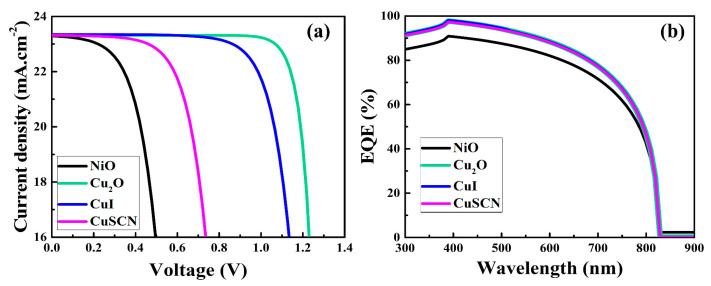
The J-V curves (**a**) and EQEs (**b**) of the as-simulated PSCs.

**Figure 3 nanomaterials-12-03315-f003:**
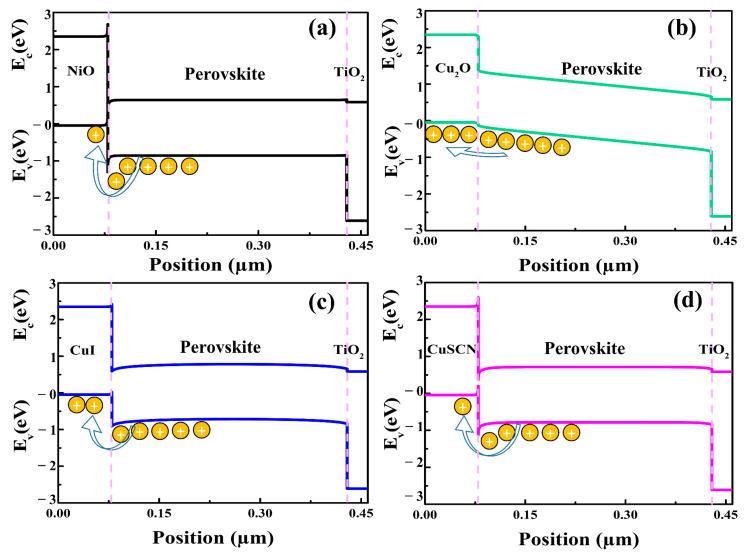
Energy band diagrams of the as-simulated PSCs with indication of energy barriers at the interfaces of different HTMs and the perovskite layer. NiO/perovskite (**a**), Cu_2_O/perovskite (**b**), CuI/perovskite (**c**), and CuSCN/perovskite (**d**).

**Figure 4 nanomaterials-12-03315-f004:**
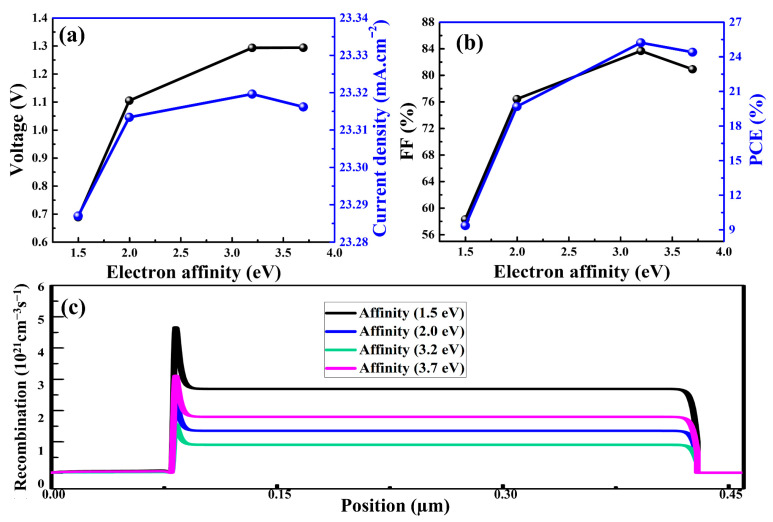
Photovoltaic parameters (**a**,**b**) and charge carrier recombination rates (**c**) of the as-simulated PSCs as a function of electron affinity of Cu_2_O.

**Figure 5 nanomaterials-12-03315-f005:**
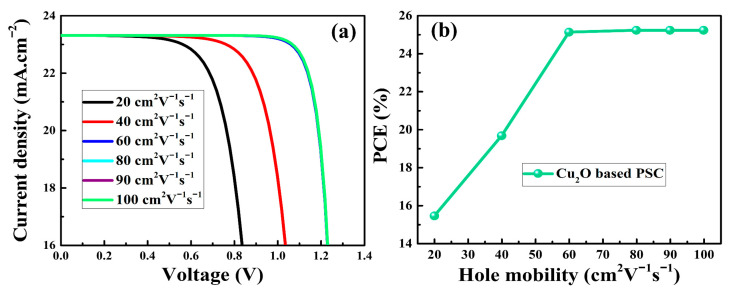
The J-V curves (**a**) and PCE (**b**) of the as-simulated PSCs as a function of hole mobility of Cu_2_O HTM.

**Figure 6 nanomaterials-12-03315-f006:**
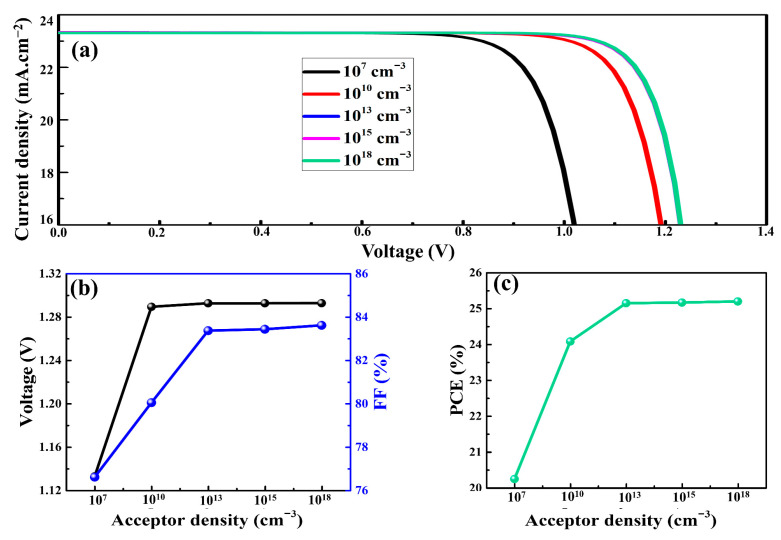
The J-V curves (**a**), V_oc_, FF (**b**), and PCE (**c**) of the as-simulated PSCs.

**Figure 7 nanomaterials-12-03315-f007:**
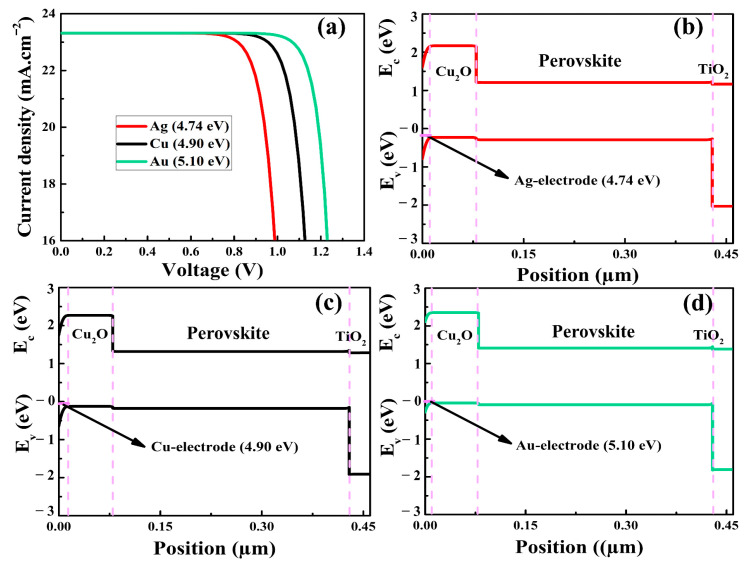
The J-V curves (**a**) and energy band alignments (**b**–**d**) of the as-simulated PSCs as a function of back electrode work function.

**Table 3 nanomaterials-12-03315-t003:** Photovoltaic parameters obtained from the simulation of PSCs with different HTMs.

Parameters	NiO	Cu_2_O	CuI	CuSCN
*V_oc_* (V)	0.720859	1.292878	1.138577	0.956405
*J_sc_* (mA·cm^−2^)	23.28859840	23.30958402	23.33917732	23.30569402
FF (%)	50.8580	83.6563	72.0141	59.9157
PCE (%)	8.5379	25.2110	21.8071	13.3550

## Data Availability

All the data presented in the manuscript can be obtained from the corresponding author by reasonable request.

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
