# Peer review of "Guidelines for Fabricating Highly Efficient Perovskite Solar Cells with Cu2O as the Hole Transport Material"

_nanomaterials, 2022, doi:10.3390/nano12193315_

Round 1

Reviewer 1 Report

In this manuscript, the authors conducted simulation investigations using SCAPS to study the effect of inorganic HTL on the photovoltaic performance of perovskite solar cells, including NiO, Cu2O, CuI, and CuSCN. The authors found that PCE significantly relies on electron affinity, hole mobility, doping density of Cu2O, and the work-function of the back-electrode, which can deliver PCE as high as 25.2%. In summary, this is an interesting work with comprehensive study and analysis. I recommend its publication in ACS Energy Letters after addressing the following questions.

1.        The authors demonstrate several drawbacks of the most used organic HTL. The inorganic HTL indeed possesses the advantage in materials cost. However, the deposition of organic HTLs is normally much simpler compared to the inorganic ones, which should also be mentioned in the introduction.

2.        After the systematical screening, the Cu2O is shown the best performance. However, Cu2O should be easy to be oxidized, which should also be considered. And can the authors predict the annealing temperature required to achieve the peripeties used for simulation in this study?

3.        Should the “Capture cross section for electrons and holes in donor tail states” be “Capture cross section for electrons and holes in acceptor tail states”?

4.        The parameters of the active perovskite layer should also be provided.

5.        The simulation results of the device with Cu2O as HTL are very different from the result reported in ref.22. The authors can give more analysis.

Author Response

Dear Myrilla Xiao,

We are very thankful for your kind letter and decision on the Manuscript "Guidelines for fabricating highly efficient perovskite solar cells with Cu2O as the hole transport material". According to the reviewer’s comments and suggestions, we made the necessary changes to the manuscript. After modifying the manuscript, we hope that your prestigious journal will accept our manuscript.

Comments and Suggestions for Authors

In this manuscript, the authors conducted simulation investigations using SCAPS to study the effect of inorganic HTL on the photovoltaic performance of perovskite solar cells, including NiO, Cu2O, CuI, and CuSCN. The authors found that PCE significantly relies on electron affinity, hole mobility, doping density of Cu2O, and the work-function of the back-electrode, which can deliver PCE as high as 25.2%. In summary, this is an interesting work with comprehensive study and analysis. I recommend its publication in nanomaterials after addressing the following questions.

  1. The authors demonstrate several drawbacks of the most used organic HTL. The inorganic HTL indeed possesses the advantage in materials cost. However, the deposition of organic HTLs is normally much simpler compared to the inorganic ones, which should also be mentioned in the introduction.

Reply: We are very thankful for your valuable comments. The relevant literatures and discussion on the methods to deposit inorganic HTMs are presented in the revised manuscript, as can be seen on pages 2 and 9.  

  1. After the systematical screening, the Cu2O is shown the best performance. However, Cu2O should be easy to be oxidized, which should also be considered. And can the authors predict the annealing temperature required to achieve the properties used for simulation in this study?

Reply: Thank you for your comments. According to the literature, the Cu2O is more stable than spiro-OMeTAD, as mentioned in the reference https://doi.org/10.1002/advs.201801169 and https://doi.org/10.1002/cssc.201901430. Therefore, Cu2O is better alternative when comparing its performance and stability to other counter parts. Furthermore, the encapsulation of the device could be another way to improve the stability of the Cu2O-based PSCs. The annealing temperature depends on the deposition methods, for the solution-processed Cu2O HTMs in the regular PSCs, for example, the annealing temperature varies from 50-60 0C for 4-6 minutes, as reported in https://doi.org/10.1002/cssc.201901430, https://doi.org/10.1002/advs.201801169.

  1. Should the “Capture cross section for electrons and holes in donor tail states” be “Capture cross section for electrons and holes in acceptor tail states”?

Reply: Thank you very much for mentioning this point. In fact the word “donor” was repeated, which is corrected in the revised manuscript. As highlighted in Table 2, column 1, the one is “Capture cross-section for electrons and holes in donor tail states” and another is “Capture cross-section for electrons and holes in acceptor tail states”.

  1. The parameters of the active perovskite layer should also be provided.

Reply: The parameters of the active perovskite layer are provided in column no.2 of Table 1.

  1. The simulation results of the device with Cu2O as HTL are very different from the result reported in ref.22. The authors can give more analysis.

Reply: Since our results are based on the simulation, which are provided for the guidance to further improve the efficiencies of PSCs with Cu2O by modulating its electronic features. The mentioned ref.22 was putted in the wrong place. In the revised manuscript the relevant references are ref.29 and ref.30 (https://doi.org/10.1002/cssc.201901430 and https://doi.org/10.1002/advs.201801169), which were used for the possible experimental approaches.

Reviewer 2 Report

The submitted paper “Guidelines for fabricating highly efficient perovskite solar cells with Cu2O as the hole transport material” by S. Sajid, S. Alzahmi, I. Ben Salem, I.M. Obaidat is devoted to simulations of photovoltaic properties of solar cell structures based on MAPbI3Br films employing SCAPS code. The SCAPS code is based on numerical solution of Poisson and continuity equations for 1D stack of materials and is widely used for simulations of solar cell structures (a proper reference to the developers would be expected in the text). The authors performed a series of calculations using different input parameters within the functionality provided by the code. The input parameters for bulk materials and for the interface regions with interface-induced electronic states were taken basically from previous works. In my opinion the title is too pretentious. In fact the authors have just checked that the set of parameters of Cu2O (tables 1 and 2; part of parameters are obviously difficult to measure and control experimentally and a high dispersion can be expected) is prospective for obtaining high PCE (25.2 %) in solar cells. It could be said that a “guideline“ is provided if the ways to control the parameters during growth of the structure were found and reported (for example, how to control the electron affinity of Cu2O). The obtained results expectedly qualitatively confirm that contacts with low barriers for holes at HTM side and for electrons at ETM side or high carrier mobility promote the high efficiency of solar cell structure. The text contains several misprints:

1. “Valance” in lines 256 and 258 should be changed to “valence”;

2. “Deices” in line 112 should be changed to “devices”.

Also the quality/resolution of the pictures is too low.

In general, in my opinion, the submitted paper after some corrections should be resubmitted to another journal. The obtained SCAPS simulation results must be supported by experimental realization for publication in Nanomaterials.   

Author Response

Dear Myrilla Xiao,

We are very thankful for your time and consideration. We modified our manuscript according to the reviewer’s comments and suggestions.

Comments and Suggestions for Authors

The submitted paper “Guidelines for fabricating highly efficient perovskite solar cells with Cu2O as the hole transport material” by S. Sajid, S. Alzahmi, I. Ben Salem, I.M. Obaidat is devoted to simulations of photovoltaic properties of solar cell structures based on MAPbI3Br films employing SCAPS code. The SCAPS code is based on numerical solution of Poisson and continuity equations for 1D stack of materials and is widely used for simulations of solar cell structures (a proper reference to the developers would be expected in the text). The authors performed a series of calculations using different input parameters within the functionality provided by the code. The input parameters for bulk materials and for the interface regions with interface-induced electronic states were taken basically from previous works. In my opinion the title is too pretentious. In fact the authors have just checked that the set of parameters of Cu2O (tables 1 and 2; part of parameters are obviously difficult to measure and control experimentally and a high dispersion can be expected) is prospective for obtaining high PCE (25.2 %) in solar cells. It could be said that a “guideline“is provided if the ways to control the parameters during growth of the structure were found and reported (for example, how to control the electron affinity of Cu2O). The obtained results expectedly qualitatively confirm that contacts with low barriers for holes at HTM side and for electrons at ETM side or high carrier mobility promote the high efficiency of solar cell structure. The text contains several misprints:

  1. “Valance” in lines 256 and 258 should be changed to “valence”;
  2. “Deices” in line 112 should be changed to “devices”.

Also the quality/resolution of the pictures is too low.

In general, in my opinion, the submitted paper after some corrections should be resubmitted to another journal. The obtained SCAPS simulation results must be supported by experimental realization for publication in Nanomaterials. 

Reply: We are very thankful for your valuable comments. The references of the developer are provided in the revised manuscript on page 2, line 83 and 84. The Cu2O shown convincing performance in PSCs (as cited in, https://doi.org/10.1002/advs.201801169, https://doi.org/10.1002/cssc.201901430), which can be further improved by altering its electrical characteristics. Therefore, our manuscript offers a few potential approaches that can be used to determine which parameters affect the performance of the devices and can be used to enhance the PCEs of the standard PSCs based on inorganic HTMs without conducting a lot of experiments that would be a waste of time, money, and energy. The word “valance” is changed to “valence” and “deices” to “devices” in the revised manuscript, as can be seen on page 9, line 250 and page 3, line 118. In addition, the figures are provided in high-quality in the revised manuscript. The revised manuscript, in our opinion, will be a valuable addition to the nanomaterials. Once again thank you so much for your time and consideration.

Reviewer 3 Report

Obaidat et al. submitted the article titled “Guidelines for fabricating highly efficient perovskite solar cells with Cu2O as the hole transport material” in Nanomaterials. A simulation investigation using the solar cell capacitance simulator (SCAPS) is carried out on the performance of regular PSCs using inorganic HTMs. Several inorganic HTMs, such as NiO, Cu2O, CuI, and CuSCN, are incorporated into PSCs to explore matching HTM that can add to the improvement in PCE. The Cu2O-based PSCs with an efficiency of more than 25% are selected as the best replacement for other counterparts in the simulation results. Please address the following comments.  

The text in the abstract between lines 23-27 seems redundant. For example, the factors like electron affinity, work function, etc should be mentioned once. Please carefully revise the abstract. In addition, the authors mentioned in the abstract that 25% increment in PCE Cu2O-based PSCs, but there is no reason provided to support this conclusion. I believe the abstract needs complete revision.

An introduction is started with the word ‘PSC’ while I recommend every abbreviation should be defined at its first use. Please check other terminologies as well such as PCE, HTM, CuI, NiO, etc.

What are the undesirable features of organic HTMs besides cost (line 43) that limit the commercialization of PSCs? There is a need to provide some examples.  Similarly, what are the desirable properties of HTMs (line 45)? There is a repetition of the word ‘desirable’.

The authors used 1-sun simulation conditions for the evaluation. However, under low-intensity lights, the behavior and properties of transport layers are also varied. I suggest the authors to study and include the following studies in a suitable position. (https://doi.org/10.1016/j.apsusc.2021.150852; https://doi.org/10.1016/j.apsusc.2020.146840; https://doi.org/10.1016/j.jmrt.2021.12.086)

Line 66, I believe the statement about firm belief is exaugurated. Please provide some quantifiable data to support your statement.

Can the authors describe the relationship between energy levels of HTMs and hole mobility? How are these parameters related?

There is a need to provide some fabrication limitations of these HTMs. The authors can add some explanation prior to the conclusion section.

The figure's quality is poor, especially the graphs. Please improve the resolution of the graphs. Use high-quality figures. Also, manuscript requires linguistic revision.

Author Response

Dear Myrilla Xiao,

We are very thankful for your kind letter and decision on the Manuscript "Guidelines for fabricating highly efficient perovskite solar cells with Cu2O as the hole transport material". We appreciate your efforts and the reviewer’s comments to revise our manuscript. We have made the modification in the manuscript according to reviewer’s comments in the revised manuscript. We hope to accept our paper in your prestigious journal after modified manuscript.

Comments and Suggestions for Authors

Obaidat et al. submitted the article titled “Guidelines for fabricating highly efficient perovskite solar cells with Cu2O as the hole transport material” in Nanomaterials. A simulation investigation using the solar cell capacitance simulator (SCAPS) is carried out on the performance of regular PSCs using inorganic HTMs. Several inorganic HTMs, such as NiO, Cu2O, CuI, and CuSCN, are incorporated into PSCs to explore matching HTM that can add to the improvement in PCE. The Cu2O-based PSCs with an efficiency of more than 25% are selected as the best replacement for other counterparts in the simulation results. Please address the following comments.  

  1. The text in the abstract between lines 23-27 seems redundant. For example, the factors like electron affinity, work function, etc should be mentioned once. Please carefully revise the abstract. In addition, the authors mentioned in the abstract that 25% increment in PCE Cu2O-based PSCs, but there is no reason provided to support this conclusion. I believe the abstract needs complete revision.

Reply: We are very thankful to you for highlighting these points. The abstract is revised accordingly, as can be seen in the revised manuscript.

  1. An introduction is started with the word ‘PSC’ while I recommend every abbreviation should be defined at its first use. Please check other terminologies as well such as PCE, HTM, CuI, NiO, etc.

Reply: We agreed with your suggestion. All of the abbreviations are defined accordingly in the revised manuscript.

  1. What are the undesirable features of organic HTMs besides cost (line 43) that limit the commercialization of PSCs? There is a need to provide some examples.  Similarly, what are the desirable properties of HTMs (line 45)? There is a repetition of the word ‘desirable’.

Reply: The undesirable features of organic HTMs are provided in the revised manuscript with examples and literatures, as mentioned on page 2, line 45-51. In addition, the desirable features of inorganic HTMs are also included, as can be seen on page 2, line 52-54. The repeating words are corrected accordingly.

  1. The authors used 1-sun simulation conditions for the evaluation. However, under low-intensity lights, the behavior and properties of transport layers are also varied. I suggest the authors to study and include the following studies in a suitable position. (https://doi.org/10.1016/j.apsusc.2021.150852; https://doi.org/10.1016/j.apsusc.2020.146840; https://doi.org/10.1016/j.jmrt.2021.12.086)

Reply: We appreciate your thoughtful comments very much. Our simulation is based on impacts of electronic features of the Cu2O-HTMs on the performance of PSCs, which only change if we alter certain parameters like electron affinity, hole mobility, and charge density, etc. Additionally, Cu2O is used in regular PSCs designs, where the perovskite layer absorbs the majority of the input photons. As a result, altering the AM1.5G solar radiation intensity will only have an impact on the perovskite layer's behavior and characteristics, which is not the aim of this study. In order to evaluate the output parameters of the PSCs, such as J-V characteristics, EQEs, energy barriers, etc., we held the AM1.5G constant while varying the electrical parameters of the Cu2O. The mentioned articles are cited in the revised manuscript accordingly.

  1. Line 66, I believe the statement about firm belief is exaggerated. Please provide some quantifiable data to support your statement.

Reply: Thank you for mentioning this point. The statement is modified accordingly, as can be seen on page 2 in the revised manuscript.

  1. Can the authors describe the relationship between energy levels of HTMs and hole mobility? How are these parameters related?

Reply: There is actually no direct correlation between energy levels and hole mobility of the HTMs because one is responsible for collection of holes from the perovskite layer and the other is the ability to transport/conduct holes. In addition, we corrected a mistake on page 6, line 154, i.e. “high hole mobility” is changed to “low energy barrier”.

  1. There is a need to provide some fabrication limitations of these HTMs. The authors can add some explanation prior to the conclusion section.

Reply: Thank you so much for such valuable comments. The discussion on the fabrication limitations of the HTMs is provided with possible ways to deposit these HTMs in the regular designs of the PSCs, as can be seen on pages no.2 and 9.

  1. The figure's quality is poor, especially the graphs. Please improve the resolution of the graphs. Use high-quality figures. Also, manuscript requires linguistic revision.

Reply: The figures are provided in high-quality in the revised manuscript. In addition, the grammatical errors are corrected accordingly.

Round 2

Reviewer 2 Report

I think the authors have improved the text and fixed all formal drawbacks. I do think the results can be interesting for the scientific community. I still think that this “simulation-only” work is more suitable for a lower impact factor journal but the final decision is up to the Editor. Also before the publication I would ask the authors to explain what is “defect-layers at the interfaces” (ETM/perovskite and HTM/perovskite). Is it interface dipole region? And why it is that thin (0,02 Angstroms – 25 times less than Bohr radius, according to Table 2).     

Author Response

Reply: We are very thankful for your consideration and time. In fact, the SCAPs-software does not take into account the recombination of charges at the corresponding interfaces (page 3, line 103 and 104). Therefore, we need to insert the interface layers, as mentioned in Table 2. This approach will highlight the role of interface engineering in determining the performance of the device compatibility between theoretical and experimental results. We notice a mistake in the unit representation for the thickness. The unit is corrected for the thickness, which is “0.002m”, as can be seen in Table 2.  

Reviewer 3 Report

The authors have satisfactorily addressed my comments. So, no further revision is required.

Author Response

Comments and Suggestions for Authors

The authors have satisfactorily addressed my comments. So, no further revision is required.

Reply: We are very thankful to you for your consideration and time.
